# Learned Belief Search: Efficiently Improving Policies in Partially Observable Settings

## Abstract

Search is an important tool for computing effective policies in single- and multi-agent environments, and has been crucial for achieving superhuman performance in several benchmark fully and partially observable games. However, one major limitation of prior search approaches for partially observable environments is that the computational cost scales poorly with the amount of hidden information. In this paper we present *Learned Belief Search* (LBS), a computationally efficient search procedure for partially observable environments. Rather than maintaining an exact belief distribution, LBS uses an approximate auto-regressive counterfactual belief that is learned as a supervised task. In multi-agent settings, LBS uses a novel public-private model architecture for underlying policies in order to efficiently evaluate these policies during rollouts. In the benchmark domain of Hanabi, LBS obtains more than 60% of the benefit of exact search while reducing compute requirements by $35\times$, allowing it to scale to larger settings that were inaccessible to previous search methods.

## 1 Introduction

Search has been a vital component for achieving superhuman performance in a number of hard benchmark problems in AI, including Go (Silver et al., 2016; 2017; 2018), Chess (Campbell et al., 2002), Poker (Moravčík et al., 2017; Brown & Sandholm, 2017; 2019), and, more recently, self-play Hanabi (Lerer et al., 2020). Beyond achieving impressive results, the work on Hanabi and Poker are some of the few examples of search being applied in large partially observable settings. In contrast, the work on belief-space planning typically assumes a small belief space, since these methods scale poorly with the size of the belief space.

Inspired by the recent success of the SPARTA search technique in Hanabi (Lerer et al., 2020), we propose *Learned Belief Search* (LBS) a simpler and more scalable approach for policy improvement in partially observable settings, applicable whenever a model of the environment and the policies of any other agents are available at test time. Like SPARTA, the key idea is to obtain Monte Carlo estimates of the expected return for every possible action in a given action-observation history (AOH) by sampling from a belief distribution over possible states of the environment. However, LBS addresses one of the key limitations of SPARTA. Rather than requiring a sufficiently small belief space, in which we can compute and sample from an exact belief distribution, LBS samples from a learned, auto-regressive belief model which is trained via supervised learning (SL). The auto-regressive parameterization of the probabilities allows LBS to be scaled to high-dimensional state spaces, whenever these are composed as a sequence of features.

Another efficiency improvement over SPARTA is replacing the full rollouts with partial rollouts that bootstrap from a value function after a specific number of steps. While in general this value function can be trained via SL in a separate training process, this is not necessary when the blueprint (BP) was trained via RL. In these cases the RL training typically produces both a policy and an approximate value function (either for variance reduction or for value-based learning). In particular, it is common practice to train *centralized* value functions, which capture the required dependency on the sampled state even when this state cannot be observed by the agent during test time.

While LBS is a very general search method for Partially Observable Markov Decision Processes (POMDPs), our application is focused on single-agent policy improvement in Decentralized

POMDPs (Dec-POMDPs) (in our specific case, Hanabi). One additional challenge of single-agent policy improvement in Dec-POMDPs is that, unlike standard POMDPs, the Markov state $s$ of the environment is no longer sufficient for estimating the future expected return for a given AOH of the searching player. Instead, since the other players' policies also depend on their entire action-observation histories, *e.g.*, via Recurrent-Neural Networks (RNNs), only the union of Markov state $s$ and all AOHs is sufficient.

This in general makes it challenging to apply LBS, since it would require sampling entire AOHs, rather than states. However, in many Dec-POMDPs, including Hanabi, information can be split between the common-knowledge (CK) trajectory and private observations. Furthermore, commonly information is 'revealing', such that there is a mapping from the most recent private observation and the CK trajectory to the AOH for each given player. In these settings it is sufficient to track a belief over the union of private observations, rather than over trajectories, which was also exploited in SPARTA. We adapt LBS to this setting with a novel public-RNN architecture which makes replaying games from the beginning, as was done in SPARTA, unnecessary.

When applied to the benchmark problem of two player Hanabi self-play, LBS obtains around 60% of the performance improvement of exact search, while reducing compute requirements by up to $35\times$. We also successfully apply LBS to a six card version of Hanabi, where calculating the exact belief distribution would be prohibitively expensive.

## 2 RELATED WORK

### 2.1 BELIEF MODELING & PLANNING IN POMDPS

Deep RL on POMDPs typically circumvents explicit belief modeling by using a policy architecture such as an LSTM that can condition its action on its history, allowing it to implicitly operate on belief states (Hausknecht & Stone, 2015). 'Blueprint' policies used in this (and prior) work take that approach, but this approach does not permit search since search requires explicitly sampling from beliefs in order to perform rollouts.

There has been extensive prior work on learning and planning in POMDPs. Since solving for optimal policies in POMDPs is intractable for all but the smallest problems, most work focuses on approximate solutions, including offline methods to compute approximate policies as well as approximate search algorithms, although these are still typically restricted to small grid-world environments (Ross et al., 2008).

One closely related approach is the Rollout algorithm (Bertsekas & Castanon, 1999), which given an initial policy, computes Monte Carlo rollouts of the belief-space MDP assuming that this policy is played going forward, and plays the action with the highest expected value. In the POMDP setting, rollouts occur in the MDP induced by the belief states[1].

There has been some prior work on search in large POMDPs. Silver & Veness (2010) propose a method for performing Monte Carlo Tree Search in large POMDPs like Battleship and partially-observable PacMan. Instead of maintaining exact beliefs, they approximate beliefs using a particle filter with Monte Carlo updates. Roy et al. (2005) attempt to scale to large belief spaces by learning a compressed representation of the beliefs and then performing Bayesian updates in this space.

Most recently MuZero combines RL and MCTS with a learned implicit model of the environment (Schrittwieser et al., 2019). Since recurrent models can implicitly operate on belief states in partially-observed environments, MuZero in effect performs search with implicit learned beliefs as well as a learned environment model.

### 2.2 GAMES & HANABI

Search has been responsible for many breakthroughs on benchmark games. Most of these successes were achieved in fully observable games such as Backgammon (Tesauro, 1994), Chess (Campbell et al., 2002) and Go (Silver et al., 2016; 2017; 2018). More recently, belief-based search techniques

---

[1] SPARTA's single agent search uses a similar strategy in the DEC-POMDP setting, but samples states from the beliefs rather than doing rollouts directly in belief space.

have been scaled to large games, leading to breakthroughs in poker (Moravčík et al., 2017; Brown & Sandholm, 2017; 2019) and the cooperative game Hanabi (Lerer et al., 2020), as well as large improvements in Bridge (Tian et al., 2020).

There has been a growing body of work developing agents in the card game Hanabi. While early hobbyist agents codified human conventions with some search (O'Dwyer, 2019; Wu, 2018), more recent work has focused on Hanabi as a challenge problem for learning in cooperative partially-observed games (Bard et al., 2020). In the self-play setting (two copies of the agent playing together), the Bayesian Action Decoder (BAD) was the first learning agent (Foerster et al., 2019), which was improved upon by the Simplified Action Decoder (SAD) (Hu & Foerster, 2020). The state-of-the-art in Hanabi self-play is achieved by the SPARTA Monte Carlo search algorithm applied to the SAD blueprint policy (Lerer et al., 2020).

There has been recent work on ad-hoc team play and zero-shot coordination, in which agents are paired with unknown partners (Canaan et al., 2019; Walton-Rivers et al., 2017; Hu et al., 2020).

## 3  SETTING AND BACKGROUND

In this paper we consider a Dec-POMDP, in which $N$ agents each take actions $a_t^i$ at each timestep, after which the state $s_t$ updates to $s_{t+1}$ based on the conditional probability $P(s_{t+1}|s_t, \mathbf{a}_t)$ and the agents receive a (joint) reward $r(s_t, \mathbf{a_t})$, where $\mathbf{a}_t$ is the joint action of all agents.

Since the environment is partially observed each agent only obtains the observation $o_t^i = Z(s_t, i)$ from a deterministic observation function $Z$. We denote the environment trajectory as $\tau_t = \{s_0, \mathbf{a}_0, ...s_t, \mathbf{a}_t\}$ and the action-observation history (AOH) of agent $i$ as $\tau_t^i = \{o_0^i, a_0^i, ..., o_t^i, a_t^i\}$. The total forward looking return from time $t$ for a trajectory $\tau$ is $R^t(\tau) = \sum_{t' \geq t} \gamma^{t'-t} r(s_t, \mathbf{a_t})$, where $\gamma$ is an optional discount factor. Each agent chooses a policy $\pi^i(\tau^i)$ conditioned on its AOHs, with the goal that the joint policy $\pi = \{\pi^i\}$ maximises the total expected return $J_\pi = \mathbb{E}_{\tau \sim P(\tau|\pi)} R(\tau)$.

Starting from a common knowledge blueprint (BP), i.e. a predetermined policy that is known to all players, in order to perform *search* in a partially observable setting, agents will need to maintain *beliefs* about the state of the world given their observations. We define beliefs $B^i(\tau_t) = P((s_t, \{\tau_t^j\})|\tau_t^i)$, which is the probability distribution over states and AOHs, given player $i$'s private trajectory. Note that in Dec-POMDPs the beliefs must model other agents' AOHs as well as the current state, since in general the policies of other players condition on these AOHs.

In general, the domain of $B^i$ (the *range*) is extremely large, but in Dec-POMDPs with a limited amount of hidden information there is often a more compact representation. For example, in card games like Hanabi, the range consists of the unobserved cards in players' hands. SPARTA (Lerer et al., 2020) assumed that the domain was small enough to be explicitly enumerated. In our case, we assume that the beliefs are over *private features* $f^i$ that can be encoded as a sequence of tokens from some vocabulary: $f^i = \prod_j f_j^i \in \mathcal{V}$. Furthermore, it simplifies the algorithm if, as is typically the case, we can factor out these private features from $\tau$ to produce a public trajectory $\tau^{pub}$; then each $\tau^i$ can be specified by the pair $(\tau^{pub}, f^i)$ and $\tau$ by $(\tau^{pub}, f^1, ..., f^N)$.

LBS is applicable to general POMDPs, which are a natural corner case of the Dec-POMDP formalism when we set the number of players to 1.

## 4  METHOD

The starting point for our method is the single-agent search version of SPARTA (Lerer et al., 2020): Given a set of BP policies, $\pi^i$, we estimate expected returns, $Q(a^i|\tau^i)$, by sampling possible trajectories $\tau$ from a belief conditioned on $\tau^i$:

$$Q(a^i|\tau^i) = \mathbf{E}_{\tau \sim P(\tau|\tau^i)} Q(a^i|\tau) \qquad (1)$$

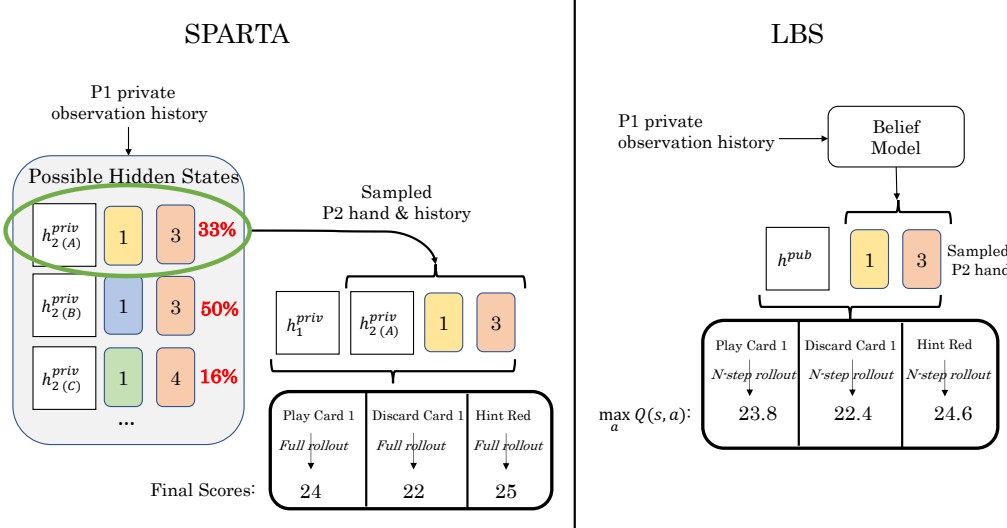

Figure 1: Comparison of SPARTA (Lerer et al., 2020) and LBS (ours). SPARTA maintains an explicit belief distribution with an accompanying AOH $h_2^{priv}$ for each belief state. LBS uses an auto-regressive belief model to sample states from the belief distribution, given the AOH. AOHs do not need to be maintained for each belief state in LBS since the model only relies on the public trajectory. Additionally, LBS uses an N-step rollout followed by a bootstrap value estimate.

Here $Q(a^i|\tau)$ is the expected return for playing action $a^i$ given the history $\tau$:

$$Q(a^i|\tau) = \mathbf{E}_{\tau' \sim P(\tau'|\tau, a^i)} R^t(\tau'), \tag{2}$$

where $R^t(\tau')$ is the Monte Carlo forward looking return from time $t$.

Whenever the argmax of $Q(a^i|\tau^i)$ exceeds the expected return of the BP, $Q(a_{BP}|\tau^i)$, by more than a threshold $\delta$, the search-player deviates from the BP and plays the argmax instead. For more details, please see Figure 1 (LHS) and the original paper.

Even though this is single-agent search, in SPARTA this process is costly: First of all, the belief $P(\tau|\tau^i)$ is an exact counterfactual belief, which requires evaluating the BP at every timestep for every possible $\tau$ to identify which of these are consistent with the actions taken by the other agents. Secondly, to estimate $R^t(\tau')$ SPARTA plays full episodes (*rollouts*) until the end of the game.

Lastly, since in Dec-POMDPs policies condition on full AOHs (typically implemented via RNNs) the games have to be replayed from the beginning for each of the sampled $\tau$ to obtain the correct hidden state, $h(\tau^i(\tau))$, for all players $i$, for each of the sampled trajectories, $\tau$.

Learned Belief Search addresses all of these issues: First of all, rather than using costly exact beliefs, we use supervised learning (SL) to train an auto-regressive belief model which predicts $P(\tau|\tau^i)$. As described in Section 3, in our setting this reduces to predicting the *private observations* for all other players $f^{-i}$, since the public trajectory is known. We use an auto-regressive model to encode the private observations as it can be decomposed to sequence of tokens $f^{-i} = \prod_j f_j^{-i}$. For scalability, as illustrated in Figure 2 (RHS), the auto-regressive belief model is parameterized by a neural network with weights $\phi$:

$$P_{exact}(f^{-i}|\tau^i) \to P_\phi(f^{-i}|\tau^i) = \prod_j P(f_j^{-i}|f_{<j}^{-i}, \tau^i). \tag{3}$$

Secondly, to avoid having to unroll episodes until the end of the game we use a learned value function to bootstrap the expected return after a predefined number, $N$, of steps. While this value function in general needs to be trained via SL, this is not necessary in our setting: Since our BPs are recurrent DQN agents that learn an approximate expected return via value-decomposition networks (VDN), we can directly use the BP to estimate the expected return, as illustrated in Figure 1 (B) :

$$R^t(\tau') \simeq \sum_{t'=t}^{t+N} r_{t'} + \sum_i Q_{BP}^i(a_{BP}^i|f_{t+N}^i, \tau_{t+N}^{pub})|\tau'. \tag{4}$$

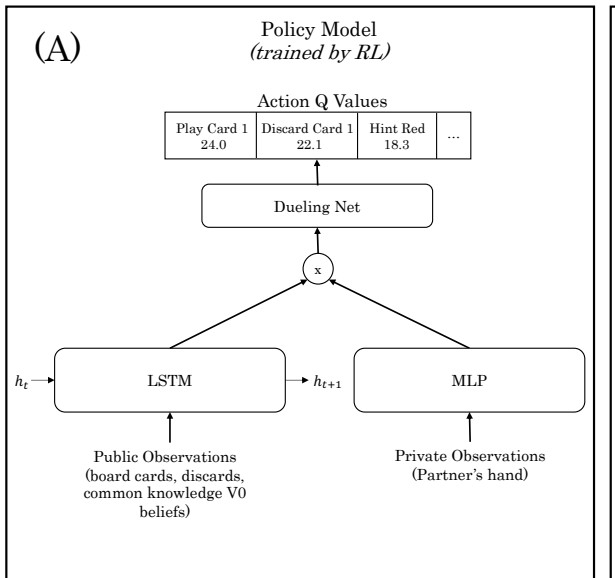
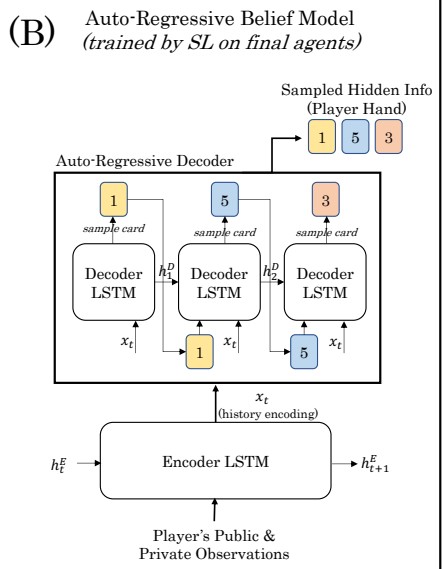

Figure 2: (**A**): Illustration of the public-LSTM network used for the BP policy. (**B**): The auto-regressive network for modeling beliefs .

This is a fairly general insight, since RL training in POMDPs commonly involves centralized value functions that correctly capture the dependency of the expected return on the central state.

Lastly, we address a challenge that is specific to Dec-POMDPs: To avoid having to re-unroll the policies for the other agents from the beginning of the game for each of the sampled $\tau$, LBS uses a specific RNN architecture. Inspired by other public-private methods (Foerster et al., 2019; Kovařík et al., 2019; Horák & Bošanský, 2019), we exploit that the public trajectory in combination with the current private observation contains the same information as the entire private trajectory and only feed the public information $\tau^{pub}$ into the RNN. The public hidden state $h(\tau_t^{pub})$ is then combined with the private observation $f_t^i$ through a feedforward neural network, as illustrated in Figure 2 (LHS).

$$\pi(\tau_t^i) \rightarrow \pi(h(\tau_t^{pub}), f_t^i) \tag{5}$$

We note that whenever it is possible to factor out the public trajectory, this architecture can be used. If not, LBS can still be used, but instead for each sampled $f^i$ we would need to reconstruct the entire game from scratch to obtain the correct model state.

We also point out that in this paper we only consider single player search where all others act according to the BP. Carrying out search for more than one player would not be theoretically sound because the trained belief model would no longer be accurate. Crucially, the learned belief model can only provide accurate implicit beliefs when the other players are playing according to the blueprint policy, because the belief model is trained before any agent conducts search. As a consequence, doing independent LBS for multiple players at the same time would lead to inaccurate beliefs and hence could lower the performance Further details, including specific architectures, of our three innovations are included in Section 5.

## 5    EXPERIMENTAL SETUP

We evaluate our methods in Hanabi, a partially observable fully cooperative multi-step game. In Hanabi, there are 5 different colors and 5 different ranks. Each card may have one or more copies in the deck. All players are of the same team trying to finish 5 stacks of cards, one for each color. They need to play the cards in order from 1 to 5 as in Solitaire. The main challenge in Hanabi is that each player can only see other players' cards, not their own. Players have to use a limited number of hint tokens to reveal information for their partners and form conventions or use theory of mind reasoning to convey information more efficiently. Please refer to Appendix B for more details on Hanabi. For simplicity we focus on 2-player Hanabi for all our experiments and note that it is straightforward to extend our method to any number of players.

## 5.1 BLUEPRINT TRAINING

As explained in Section 3 the public and private observations in Hanabi can be factorized into public and private features. We modify the open-source Hanabi Learning Environment (HLE) (Bard et al., 2020) to implement this. Here, the only private observation is the partner's hand, while all other information is public. There are many different options for implementing the public-RNN concept. We end up with the design shown in Figure 2 (A). Specifically, the LSTM only takes public features as input while an additional MLP takes in the concatenation of private and public features. The outputs of the two streams are fused through element-wise multiplication before feeding into the dueling architecture (Wang et al., 2016) to produce Q values. We have also experimented with other designs such as using concatenation in place of the element-wise multiplication, or feeding only private observation to the MLP. Empirically we find that the design chosen performs the best, achieving the highest score in self-play.

To train our BP with reinforcement learning, we base our implementation on the open-source code of Other-Play (Hu et al., 2020), which includes several recent advancements for RL in Hanabi. We follow most of their practices such as value decomposition network, color shuffling and auxiliary tasks. We also leave their hyper-parameters unchanged. For completeness, we include a detailed description of experimental setup and hyper-parameters in Appendix C.

## 5.2 BELIEF LEARNING

The belief model is trained to predict the player's own hand given their action observation history. An overview of the architecture is shown in the right panel of Fig 2. An encoder LSTM converts the sequence of observations to a context vector. The model then predicts its own hand in an auto-regressive fashion from oldest card to newest. The input at each step is the concatenation of the context vector and the embedding of the last predicted card. The model is trained end-to-end with maximum likelihood:

$$\mathcal{L}(c_{1:n}|\tau) = -\sum_{i=1}^{n} \log p(c_i|\tau, c_{1:i-1}), \tag{6}$$

where $n$ is the number of cards in hand and $c_i$ is the $i$-th card.

We use a setup similar to that of reinforcement learning to train the belief model instead of a more traditional way of creating a fixed train, test and validation set. We use a trained policy and a large number of parallel Hanabi simulators to continuously collect trajectories and write them into a replay buffer. In parallel we sample from the replay buffer to train the belief model using a supervised loss. This helps us easily avoid over-fitting without manually tuning hyper-parameters and regularization. The RL policy used to generate data is fixed during the entire process.

## 5.3 LBS IMPLEMENTATION DETAILS

Learned Belief Search is straightforward once we have trained a BP and a belief model. The search player samples hands from the belief model and filters out the ones that are inconsistent with current game status based on their private observation. In the extremely rare case where the belief model fails to produce a sufficient number of legal hands, it reverts back to the BP. To understand the impact of various design parameters, we experimented with both playing out all trajectories until the end of the game, LBS-inf, as well as rolling out for a fixed number of steps, LBS-k and using a bootstrapped value estimate at the final state. Similar to SPARTA, the search actor only deviates from the BP if the expected value of the action chosen by search is $\delta = 0.05$ higher than that of the BP and a UCB-like pruning method is used to reduce the number of samples required.

## 6 RESULTS

**Belief Quality**. We first examine the quality of the learned belief model by looking at its cross entropy loss for predicting hidden card values. We compare against two benchmarks: the exact beliefs marginalized over each card, and an auto-regressive belief based only on grounded information. The grounded belief predicts a distribution over card values proportional to remaining card counts, for all card values consistent with past hints. A formal definition is provided in Appendix A.

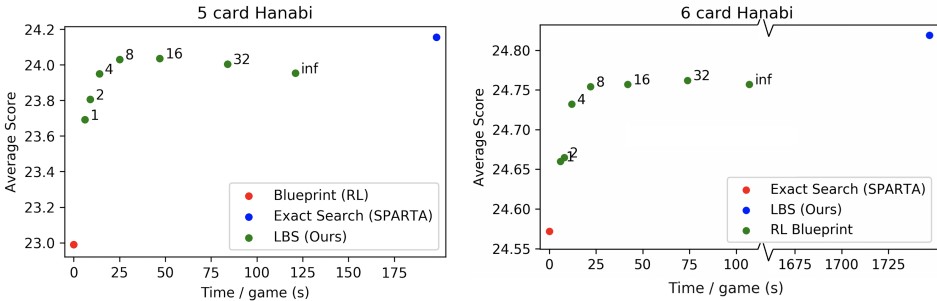

Figure 3: Comparison of speed and average score of different policies in 5-card (**left**) and 6-card (**right**) Hanabi. The number next to each LBS data point is the rollout depth. LBS with different rollout depths provide a tunable tradeoff between speed and performance. LBS provides most of the performance benefits of exact search (SPARTA) at a fraction of the compute cost, and the speedup grows with the size of the belief space (5-card vs 6-card).

| Method | Time | RL-1H | RL-5H | RL-10H | RL-20H |
|---|---|---|---|---|---|
| Blueprint | *<1s* | 15.38 | 22.99 | 23.70 | 24.08 |
| SPARTA | *215s* | 19.53 | 24.16 | 24.39 | 24.52 |
| LBS-inf | *121s* | 18.88 | 23.95 | 24.29 | 24.42 |
| LBS-32 | *84s* | 19.05 | 24.01 | 24.31 | 24.45 |
| LBS-16 | *47s* | 19.27 | 24.04 | 24.29 | 24.48 |
| LBS-8 | *25s* | 19.14 | 24.03 | 24.28 | 24.43 |
| LBS-1 | *6s* | 17.99 | 23.69 | 24.06 | 24.26 |

Table 1: Average scores in 2-player Hanabi with different search variants. Time column shows the average wall-clock time of each method to play a game. Columns of RL-$k$H show performance of different methods applied on blueprint policies trained for different amounts of time. Each cell contains the mean averaged over 5000 games. Please refer to Appendix D for the complete version of the table that includes the standard error of mean, percentage of perfect games, and more test scenarios. LBS variants achieve a substantial fraction of the policy improvements of SPARTA over the blueprint at a lower computational cost.

We generate 1000 games with our trained RL policy and compute the two benchmarks together with the loss of the learned belief (Eq. 6) from a fully trained belief model. Figure 4 shows how these 3 values change over the course of games. We see that our learned belief model performs considerably better than the grounded belief. There is still a gap between learned belief and exact belief, especially in the later stage of the game. More powerful models such as transformers (Vaswani et al., 2017) may further improve the belief learning but we leave it for future work.

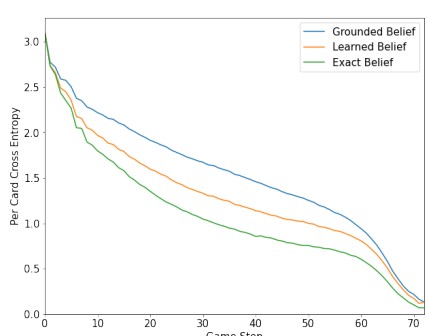

Figure 4: Per-card cross entropy with the true hand for different beliefs in games played by BP.

**Performance**. Figure 3 summarizes the speed/performance tradeoffs of LBS compared to running an RL policy directly or using an exact search method (SPARTA). In 5-card Hanabi (left), we start with a blueprint policy that achieves an average score of 22.99 after 5 hours of training, and apply LBS with different rollout lengths as well as an exact search. LBS achieves most of the score improvement of exact search while being much faster to execute. Changing the rollout depth allows for tuning speed vs. performance. As we move to a larger game (6-card Hanabi), SPARTA

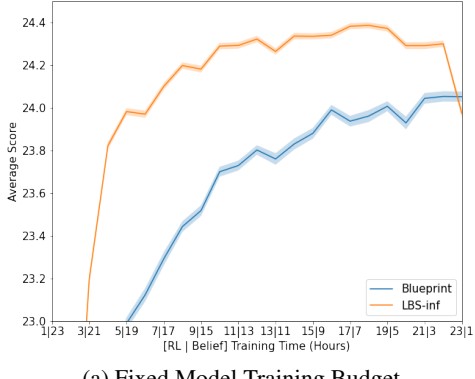

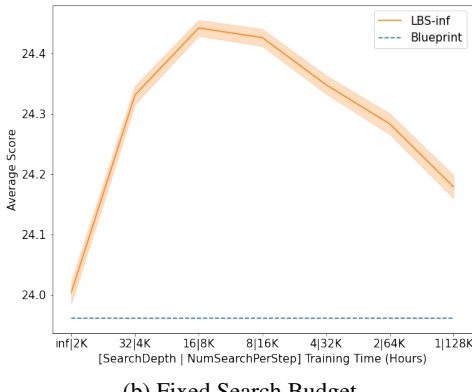

(a) Fixed Model Training Budget (b) Fixed Search Budget

Figure 5: (a) Result of fixed budget (24 Hours) at training time. The ticks "$a|b$"on x-axis means $a$ hours to train BP and $b$ hours to train belief model. (b) Result of fixed budget at test time. The ticks "$a|b$" on x-axis means run search for $a$ steps before bootstrapping with Q function and run $b$ search per move. Each data point on both figures is evaluated on 5000 games and the shaded area is the standard error of mean.

becomes 10x slower while LBS runs at approximately the same speed and continues to provide a similar fraction of the performance improvement.

A more comprehensive analysis of these tradeoffs is shown in Table 1. We train the BP with RL for 40 hours in total and use 5 snapshots dubbed **RL-$k$H** which have been trained for 1, 5, 10, 20 and 40 hours to showcase the performance of our method given different BPs of different strength. Note that in Table 1 we only show results on 4 snapshots for conciseness. A more complete version of the table with standard error of mean (s.e.m.) and more data points can also be found in Table 4 in Appendix D. The belief models are trained to convergence for each BP. For comparison, we rerun SPARTA on our models which can be seen as a expensive upper bound for LBS. All search methods run 10K rollouts per step. The best performing LBS variant is LBS-16, delivering 86% (on RL-10H) to 94% (on RL-1H) of the improvement comparing to the exact search while being $4.6\times$ faster. Even the cheapest method, LBS-1, returns a decent improvement of 45% on average and $35.8\times$ speedup. It also worth noting that LBS performs strongly even on a quite poor RL policy, RL-1H, demonstrating high versatility of the method.

We note that the more expensive LBS-inf is not the best performing one, it consistently underperforms some of the LBS-k versions by a small, in many cases significant, amount. We hypothesis that under LBS-inf it is more likely for the agents to reach a state that is under-explored during training. Therefore the approximate belief will be less accurate and the estimate Q-value be wrong. The LBS-k method where the Q-value after k steps is as a bootstrap may naturally avoid those situations since the BP may also have low Q values for under-explored states. One piece of evidence for this theory is that in LBS-inf, 0.1% of the games end up with a completely failed belief prediction and have to revert back to BP while the same only happens to 0.0083% of the games for LBS-k.

Clearly, this is a potential problem: part of the reason search works well is that it can discover moves that were underestimated and, consequently, under-explored by the BP. The brittleness of the learned belief, in combination with the difference in overall belief quality (Fig 4), help explain the difference in performance between LBS and exact methods like SPARTA.

**Fixed Budget Training & Testing**. Since one of the motivations for LBS is speed, it would be interesting to know how we could allocate resources at training and test time to maximize performance given a fixed computational budget. For fixed training budget, we train the BP for $l$ hours and then train belief model for $24 - l$ hours. We evaluate these combinations with LBS-inf as well as BPs themselves. As shown in Figure 5a, with longer RL training, the BP improves monotonically, but the final performance suffers due to a poor belief model. The best combination is $\sim$18 hours for RL and $\sim$6 hours for belief learning.

We then take the models from the best combination to study how to allocate compute between the number of rollouts and rollout depth. We start with LBS-1 and 128K search per step, and then halve the number of searches as we double the search depth. The result is shown in Figure 5b. If we compare the results here with those from Table 1, we see that although LBS-16 still has the best

|  | **Blueprint** | **SPARTA** | **LBS-32** | **LBS-16** |
|---|---|---|---|---|
| **Performance** | $24.57 \pm 0.01$ | $24.82 \pm 0.01$ | $24.76 \pm 0.01$ | $24.76 \pm 0.01$ |
|  | 75.94% | 87.80% | 84.12% | 84.06% |
| **Run Time** | $<1$s | 1747s | 74s | 42s |

Table 2: Result on 6-card Hanabi variant. Each cell contains the mean and standard error of mean over 5000 games in the first row and percentage of perfect games (25 points) in the second row.

performance, the relative strength between LBS-32 and LBS-8 flips, indicating that the trade-off may still matter in some cases.

**Extension to 6-card Hanabi**. To further demonstrate the scalability of LBS compared to SPARTA, we test them on a modified version of Hanabi where each player holds 6 cards instead of 5. This is not an official variant of Hanabi and it is easier to achieve a high score than the standard form due to more choices at each turn and shorter episode length. We train the BP and belief model with the same method. The results are shown in Table 2. The SPARTA method runs $8\times$ slower than it does on standard Hanabi while LBS runs faster due to shorter games, delivering 76% of the improvement with $42\times$ less time.

## 7 CONCLUSION

We presented Learned Belief Search, a novel search algorithms for POMDPs that can be used to improve upon the performance of a blueprint policy at test time whenever a simulator of the environment is available. We also presented extensions of LBS that make it applicable to fully cooperative, partially observable multi-agent settings. The heart of LBS is an auto-regressive model that can be used to generate samples from an approximate belief for any given AOH.

While LBS achieves strong performance on the benchmark problem Hanabi, our work also clearly points to a number of future directions. For a start, the search process can bring the belief model to under-explored regions of the state space. This could be addressed by retraining the belief model on the data generated from LBS.

Another interesting direction for future work is to amortize the search process, *e.g.* by integrating it into the training process, and to extend LBS to multiplayer and multi-step search. To enable these directions, and many others, we plan to open-source all of the code for LBS.

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

## A   DEFINITION OF GROUNDED BELIEF AND EXACT BELIEF

The grounded belief for each card in hand is defined as

$$p(c_i = C_j | c_{1:i-1}) = \frac{\mathbb{1}(c_i, C_j) \cdot f(C_j | c_{1:i-1})}{\sum_{j=1}^{K} \mathbb{1}(c_i, C_j) \cdot f(C_j | c_{1:i-1})}, \tag{7}$$

where $C_1, ..., C_{K=25}$ are all possible outcomes for a card, $\mathbb{1}(c_i, C_j)$ is an indicator function telling whether a particular outcome $C_j$ is plausible for card $c_i$ based on the information revealed through hints, and $f$ returns number of remaining copies of a card by excluding played cards, discarded cards, partner's hand and my cards in previous slots.

On the other side of the spectrum, we can compute the exact belief over hand as in Lerer et al. (2020) by tracking all possible hands and performing counterfactual filtering as game progresses. We then compute the marginal distribution for each card in hand as:

$$p(c_i = C_j | c_{1:i-1}) = \frac{\sum_{k=1}^{N} \mathbb{1}\left(h_i^{(k)} = C_j, h_{1:i-1}^{(k)} = c_{1:i-1}\right) \cdot q(h^{(i)})}{\sum_{k=1}^{N} \mathbb{1}\left(h_{1:i-1}^{(k)} = c_{1:i-1}\right) \cdot q(h^{(i)})}, \tag{8}$$

where $q(h^{(k)})$ is the counterfactual distribution of my possible hands.

## B   DESCRIPTION OF HANABI

Hanabi is a fully cooperative partially observable multi-player card game. All players in the game work towards a common goal. The deck consists of 5 colors and 5 ranks from 1 to 5. For each color, there are 3 copies of 1s, 2 copies of 2, 3, 4s and 1 copy of 5, totaling 50 cards. At beginning of a game, each player draw 5 cards. However, they cannot observe their own cards but instead can see all of their partners' cards. Players take turn to move and the goal for the team is to play cards of each color from 1 to 5 in the correct order to get points. For example, at the beginning of the game where nothing has been played, the 1s of every color can be played. If red 1 is played, then red 2 become playable while the 1s of all other colors remain playable. Playing a wrong card will cause the entire team to lose 1 life token and the game will terminate early and the team will get 0 point if all 3 life tokens are exhausted. At each turn, the active player can either play a card, discard a card, or select a partner and reveal information about their cards. To reveal information, the active player can either choose a color or a rank and *every* card of that color/rank will be revealed to the chosen player. The game starts with 8 information tokens. Each reveal action costs 1 token and each discard action regains 1 token. Players will draw a new card after play/discard action until the deck is finished. After the entire deck is drawn, each player has one last turn before the game ends.

## C   ADDITIONAL DETAILS ON EXPERIMENTAL SETUP

As mentioned in the main text, we base our experiments on the existing implementations of Other-Play and SPARTA for training blueprint and running search respectively and therefore inherit many of their hyper-parameters. For completeness we include a detailed documentation of the experimental setup here.

The blueprint policy training is set up in a distributed fashion. On the simulation side, 6400 Hanabi simulators run in parallel and the observations produced by those simulators are batched together for efficient neural network computation on GPU. Due to the high computational efficiency of the Hanabi-Learning-Environment (Bard et al., 2020) and an effective batching technique introduced in Hu & Foerster (2020), a single machine with 40 CPU cores and 2 GPUs is sufficient to run entire simulation workload. The simulated trajectories are collected into a prioritized replay buffer Schaul et al. (2015). The priority of each trajectory is computed as $\xi = 0.9 * \max_i \xi_i + 0.1 * \bar{\xi}$ (Kapturowski et al., 2019) where $\xi_i$ is the per step absolute TD error. On the training front, we sample mini-batches from the replay buffer to update the policy. The simulation policy is synced with the policy being trained every 10 gradient steps. The exploration is handled by epsilon greedy method. At the beginning of each simulated game, each player samples an epsilon independently from $\epsilon_i =$

| Hyper-parameters | Value |
|---|---|
| `# replay buffer related` | |
| `burn_in_frames` | 5000 |
| `replay_buffer_size` | 131072 ($2^{17}$) |
| `priority_exponent` | 0.9 |
| `priority_weight` | 0.6 |
| `max_trajectory_length` | 80 |
| `# optimization` | |
| `optimizer` | Adam (Kingma & Ba, 2014) |
| `lr` | 6.25e-05 |
| `eps` | 1.5e-05 |
| `grad_clip` | 5 |
| `batchsize` | 64 |
| `# Q learning` | |
| `n_step` | 3 |
| `discount_factor` | 0.999 |
| `target_network_sync_interval` | 2500 |

Table 3: Hyper-Parameters for Reinforcement Learning

$\alpha^{1+\beta*\frac{i}{N-1}}$ where $\alpha = 0.1$, $\beta = 7$ and $N = 80$. The rest of the hyper-parameters can be found in Table 3.

The belief model is trained under a similar setting as blueprint policy with 3 major modifications. Firstly we load a pretrained policy for running simulations and keep it fixed. Secondly the trajectories are drawn uniformly from the replay buffer with no priorities. Finally the model is trained with the auto-regressive maximum likelihood loss defined in Eq. 6 with supervised learning. The belief model is trained with Adam optimizer with learning rate $2.5 \times 10^{-4}$ and `eps` $= 1.5 \times 10^{-5}$.

The search experiments are benchmarked under the resource constraint of 10 CPU cores and 1 GPU. At each step, we perform around 10K searches evenly distributed across legal actions. Searches run parallel over multiple CPU cores while neural network computations are batched across searches for maximum efficiency. The search will only change the move if the expected value of the action chosen by search is $\delta = 0.05$ higher than that chosen by the BP. A UCB-like pruning method is used to reduce the number of samples required.

# D    FULL LEARNED BELIEF SEARCH RESULT

| Test Method & Run Time | RL-1H | RL-5H | RL-10H | RL-20H | RL-40H |
|---|---|---|---|---|---|
| Blueprint *<1s* | $15.38 \pm 0.05$ 0.02% | $22.99 \pm 0.03$ 22.38% | $23.70 \pm 0.01$ 40.12% | $24.08 \pm 0.01$ 53.90% | $24.29 \pm 0.01$ 63.30% |
| SPARTA *215s* | $19.53 \pm 0.03$ 0.03% | $24.16 \pm 0.02$ 48.90% | $24.39 \pm 0.01$ 61.14% | $24.52 \pm 0.01$ 69.9% | $24.58 \pm 0.01$ 73.16% |
| LBS-inf *121s* | $18.88 \pm 0.03$ 0.04% | $23.95 \pm 0.02$ 42.16% | $24.29 \pm 0.01$ 57.10% | $24.42 \pm 0.01$ 65.14% | $24.52 \pm 0.01$ 71.12% |
| LBS-32 *84s* | $19.05 \pm 0.03$ 0.02% | $24.01 \pm 0.02$ 42.60% | $24.31 \pm 0.01$ 58.82% | $24.45 \pm 0.01$ 67.56% | $24.53 \pm 0.01$ 71.78% |
| LBS-16 *47s* | $19.27 \pm 0.03$ 0.02% | $24.04 \pm 0.02$ 45.40% | $24.29 \pm 0.02$ 57.98% | $24.48 \pm 0.01$ 68.18% | $24.54 \pm 0.01$ 72.14% |
| LBS-8 *25s* | $19.14 \pm 0.03$ 0.08% | $24.03 \pm 0.02$ 44.64% | $24.28 \pm 0.02$ 57.32% | $24.43 \pm 0.01$ 66.50% | $24.49 \pm 0.01$ 68.78% |
| LBS-4 *14s* | $18.75 \pm 0.03$ 0.06% | $23.95 \pm 0.02$ 41.56% | $24.26 \pm 0.02$ 55.38% | $24.41 \pm 0.01$ 64.50% | $24.45 \pm 0.01$ 66.40% |
| LBS-2 *9s* | $18.26 \pm 0.04$ 0.02% | $23.81 \pm 0.02$ 36.80% | $24.12 \pm 0.01$ 50.60% | $24.35 \pm 0.01$ 61.16% | $24.38 \pm 0.01$ 63.88% |
| LBS-1 *6s* | $17.99 \pm 0.04$ 0.00% | $23.69 \pm 0.02$ 32.90% | $24.06 \pm 0.02$ 48.66% | $24.26 \pm 0.02$ 58.36% | $24.37 \pm 0.01$ 62.72% |

Table 4: Learned Belief Search Result. RL-kH means blueprint policy after k hours of training with RL. The numbers beneath the method name in the leftmost column is the average run time at test time. Each cell contains the mean and standard error of mean over 2000 games in the first row and percentage of perfect games (25 points) in the second row.

