# OpenReview forum: "Learned Belief Search: Efficiently Improving Policies in Partially Observable Settings"
_ICLR.cc/2021/Conference — Reject_

### Official Review · AnonReviewer3 · 2020-10-28
**Interesting work on POMDPs but there is room for improvement**

**Rating:** 5
**Confidence:** 4

**Review:**

Summary:

This work provides a novel extension to the state-of-art approach (SPARTA) to solving Dec-POMDPs, a co-operative variant of partially observable stochastic games. Its main idea relies on the segmentation of action-observation histories (AOH) into more manageable public-private factors and training belief models to predict the belief on a trajecotory. Specifically, this work examines a single Dec-POMDP setting - the co-operative card-game Hanabi with two players.

##########################################################################

Pros:
Well pitched experiment, especially the use of snap-shotting to show improvement over time. Further, the inclusion of fixed budget analysis strongly emphasizes the utility of the method.

Using a natural fit of deep learning for modeling belief histories and trading-off exactness for speed, the work demonstrates a clear computational advantage over existing state-of-art approach.

##########################################################################

Cons:
More detail is required to explain why multi-agent search is not theoretically sound.  If the trained model is only accurate for single agent search, is it not possible to train a model that is compatible with multi-agent search under different settings for max range?
SPARTA shows that multi-agent search provides substantial improvements to the final scores at a large computational expense. Your work seems like it should be able to balance a trade-off for both of these functions.

For simplicity, the authors focus on 2-player Hanabi and claim it to be straightforward to extend to any number of players. I think this was a mistake, SPARTA itself was computationally limited to single-agent search for these 3-player and up variants. Applying your approach in these experiments would make a stronger case for elevation of the state-of-the-art than the 6-card variant.

The experimental setup requires significantly more details on the hardware used for training, testing and validating.

##########################################################################

Questions during rebuttal period:

In addition to the questions raised in the cons above, I have the following questions:
1. Have you examined how this would look for more general partially observable settings? In particular, work on POSGs with public observations [Horák, K.; and Bosanský, B. 2019. Solving Partially Observable Stochastic Games with Public Observations. In AAAI Conf. on Artificial Intelligence, 2029–2036. AAAI Press.].
2. Are there any other DEC-POMDP settings that you considered? It may be insightful to compare the effects of different feature spaces.

#########################################################################
Additional Comments/Suggestions

The x-axes in Figures 4a & 4b while understandable could be formatted in a formatted that does not require an extremely detailed reading of the Results section, for example in Fig 4a you could specify that the training budget was 24 hours and then use percentage of the training time spent on RL (from 5% up to 95%, which would be equivalent to 1|23 and 23|1).

Table 1 is very busy, it may be worthwhile either taking fewer snapshots (or fewer rows) or removing the standard error of mean just to make it more readable. Especially since you include the full listing at Table 3.

Some typos and errors:
Missing reference for SAD in Section 2.2

##########################################################################

Reasons for score:
Overall the approach of using a learned belief model to speed up costly belief searching is interesting. My main concern is on the soundness claim that limits the experimental scope to single-agent search. However, even with a comparison on just single-agent search this work is a fine fit for ICLR.

---

> ### Author Response · Authors · 2020-11-20
> **Response**
>
> Thanks for the insightful review. Here are our response.
>
> **Regarding multi-player search:** There is a fundamental difference between the belief computation used in SPARTA and in LBS: SPARTA uses an *exact* counterfactual belief, which means that the blueprint policy is evaluated for every possible trajectory and compared to the action taken. In contrast, in LBS this process is done by learning a belief model which calculates the counterfactuals entirely implicitly.
>
> Crucially, the learned belief model can only provide accurate implicit beliefs when the other players are playing according to the blueprint policy, because the belief model is trained before any agent conducts search.
> As a consequence, doing independent LBS for multiple players at the same time would lead to inaccurate beliefs and hence could lower the performance. We have updated the paper to address this in more detail.
>
> Therefore, extending LBS to multi-player search is far from trivial. One potential option is to instead combine LBS with exact single-agent search in SPARTA, using the LBS policy as a starting blueprint. This would still be very expensive and the LBS agent would be using a wrong belief model, which could reduce the advantage of 2 player search.
>
> **Regarding experimental setup:** Thanks for raising this up. Originally we refer the reader to the experimental setups of prior works Other-Play & SPARTA for the details since we adopt their open sourced code for the experiments. However, we have added a section in appendix elaborate the most relevant details for completeness in our revision. Please have a look.
>
> **Regarding Question(1):** Our method is designed for fully cooperative settings. For example, we assume that the blueprint policy is common knowledge and that all other players (apart from the player doing search) play according to the blueprint. Clearly, this is not very realistic in non-cooperative settings. In particular, if it is known that a player is doing LBS based on a blueprint, this could make the searching player more exploitable. We do however take extensive advantage of the public observations. Our agents use a public-private architecture that enables us to efficiently do Monte Carlo rollouts without having to unroll histories from the beginning of the episode. Thanks for pointing us to the paper. We have cited it in the relevant place as reference.
>
> **Regarding Question(2):** Two player Hanabi is a rich and computationally challenging problem that has been thoroughly benchmarked but never *solved*. In particular, Hanabi was also used for evaluating SPARTA, which makes it easy for us to compare LBS both from a computational and performance point of view.  Lastly, while there are other Dec-POMDP benchmarks, none of them feature extremely large state spaces, and LBS is intended only for Dec-POMDPs with extremely large state spaces.
>
> **Regarding Other Comments:** Many thanks for the feedbacks on the presentation of results. We updated the Table 1 to make it more concise and also added a new plot, Figure 3, to better convey the main message of this paper.

---

### Official Review · AnonReviewer4 · 2020-10-28
**recommendation to Reject**

**Rating:** 5
**Confidence:** 4

**Review:**

Summary:

The paper proposed a computationally efficient search procedure for partially observable environments. The key idea behind the proposed method, LBS,  is to obtain Monte Carlo estimates of the expected return for every possible action in a given action-observation history of an agent. The proposed method reduced the complexity of sampling from belief space over the SOTA approach SPARTA by introducing an approximate auto-regressive counterfactual belief model over the opponent's private hands.  An empirical study on cooperative multi-agent game Hanabi shows that LBS could achieve considerable score while reducing compute requirements by 35x compare to previous SPARTA.
##########################################################################
Reasons for score:

Overall, I'd vote for rejection of the paper. My primary concern is that the LBS is highly dependent on the well-trained Blue Policy. It's not clear to me how the importance of BP contributes to the final performance. To be specific, would LBS fail when based on a broken BP policy(see cons below). Hopefully, the authors can address my concern in the rebuttal period.
##########################################################################
Pros:
 1.The idea that using a learned belief model to replace exact sampling is interesting in general.
2.Overall, the paper is well written. The motivation of the proposed method is clear and sound. As an improvement over SPARTA, LBS has addressed one of the main issues of scalability. LBS offered a choice to apply SPARTA to more complex environments.


##########################################################################
Cons:

1. The key concern about the paper is that the choice of BP, which decides the performance of the belief model. It is natural that LBS will work If the belief model is perfect. But not vice versa. There is no guarantee that LBS would work if the belief model is broken or even flawed.
a).How does LBS perform when using an unconverging learned belief model? Or value model?
b).It is not clear to me that the choice of the number of factorized private features.
2.Although LBS provides several ablation studies of the choice of BP and Run time, but I'm not convinced of the efficacy of LBS, especially in more complex environments, e.g., Contract Bridge.

##########################################################################
Questions during the rebuttal period:

Please address and clarify the cons above

#########################################################################
Others
Reference alignment: Figure 1 (LHS), Figure 1(A), Fig 2.

---

> ### Author Response · Authors · 2020-11-20
> **Response**
>
> We would like to thank the reviewer for their valuable feedbacks. Below are our responses to the concerns.
>
> The reviewer is correct that for a sufficiently bad blueprint policy (e.g. a random policy), LBS will not produce a good policy. However, both SPARTA and LBS are shown to produce the largest improvements when the blueprint policy is mediocre. For example, the blueprint in the RL-5H column of Table 1 achieves a score of 22.99, which is comparable to that of the hobbyist heuristic agent SmartBot (O’Dwyer 2019), but SPARTA and LBS reduce the gap from a perfect score by more than half. To further demonstrate it, we have added an additional set of experiments using an RL policy that is only trained for 1 hour, RL-1H, to Table 1. RL-1H gets merely 15.38 in selfplay but LBS can help boost the performance to 19.27, delivering 94% of the performance gain of SPARTA while being 4.6$\times$ faster.
>
> **Regarding cons (1):** The blueprint (BP) does not _decides_ the performance of the belief model. The belief model is trained to predict distribution of hidden information when our partner executes the BP policy. The quality of the belief model should only be judged by whether it reflects such distribution faithfully instead of the strength of the BP. A good belief model is definitely crucial to the performance of the LBS method. The rightmost data point of Figure 4(a), where the belief model is only trained for 1 hour, LBS underperforms the blueprint. However, while it’s not always possible to find a good blueprint policy, we can always allocate more time to belief model training.
>
> **Regarding cons (2):** It is not trivial to apply LBS directly on Contract Bridge because it is not a Dec-POMDP, i.e. not fully cooperative. We also note that while there are other Dec-POMDP benchmarks, none of them feature extremely large state spaces, and LBS is intended only for Dec-POMDPs with extremely large state spaces.

---

### Official Review · AnonReviewer1 · 2020-10-30
**Good paper to read, not convincing presentation.**

**Rating:** 5
**Confidence:** 3

**Review:**

This paper proposes a new approach to search in POMDP environments. The main idea is to extend a previous SPARTA search technique with a belief learning ability, called Learned Belief Search (LBS). While the existing work maintains explicit belief representation, LBS is an approximate auto-regressive counterfactual belief learning method that is based on supervised learning. Belief learning is showed to both have a good generalization and reduce searching time significantly. Experiments are mainly done on the benchmark problem of two-player Hanabi self-play.

The research direction is interesting and worth pursuing given recent work on this topic. Learning belief representation makes sense in this problem setting. The search would benefit a lot if computation time can be reduced through a cheap belief computation. While I found belief learning is commmon in single-agent POMDP tasks, the motivation of why LBS can be challenging and useful for DEC-POMDP or the multi-agent game domains is not convincing.

The experiment can also be improved with regards to the elaboration of experiment settings, the problem description, and discussions.
The result in Table 1 is not discussed in the main text. It's hard to understand the numbers reported in Table 1 and 3. May the authors elaborate in the response? It would also be helpful if a brief description of the Hanabi game can be included in Appendix too.  Comparisons of different design/hyperparameter choices can also help to judge the benefit of using learning for belief representation.

Minor comment:
- missing reference in page 3: "Simplified Action Decoder (SAD) (?)"

---

> ### Author Response · Authors · 2020-11-20
> **Response**
>
> Thank you for the review. We would like to clarify a bit on the challenges of LBS on DEC-POMDP. LBS is applicable to both POMDPs and DEC-POMDPs, but DEC-POMDP provides additional challenges because we need to ensure that the partner's policy can be executed conditional only on the public state. Regarding why it's useful for DEC-POMDP, see the results of SPARTA/LBS in Hanabi, where it provides huge score improvements. If the reviewer could clarify why they don't see these techniques as challenging and useful, we'd be happy to elaborate further.
>
> We agree the presentation of results can be improved. In fact, we made a typo in the main text where we should be referring to Table 1 instead of Table 3 (now the Table 4 in the revision). We have updated the Table1 to make it more concise and easy to read. In Table 1, we take 4 different blueprint policies with different strengths. They are trained with the same RL method for 1, 5, 10, 20 hours respectively. These policies are then evaluated with different methods: blueprint only, SPARTA, and LBS-k with k being the number of rollout steps. The average performance of each method & model combination is shown in the table. The ‘Time’ column tells the average running time of each method. Table 3 (Table 4 in the revision) is a more completed version of Table 1 with the standard error of mean, percentage of perfect games, and two more test scenarios for LBS-k. In addition, we also add a new plot, Figure 3, to showcase the main benefit of LBS. The discussion accompanies the new plot can be found at the beginning of the *Performance* subsection on page 7.
>
> We have a very brief introduction of Hanabi at the beginning of Section 5. We agree that a more detailed explanation could be helpful and thus have added it in the appendix. We briefly discussed different architectures for belief learning at Section 5.1. However, it is not the focus of this paper to design the best architecture for belief modeling. The benefit of belief is demonstrated in the results section in two ways. In the **belief quality** subsection, we show that the learned belief is better than the auto-regressive v0 belief computed using ground information. In the **performance** subsection we show that search using the learned belief (i.e. LBS) can achieve good trade-off between speed & performance and thus provide a scalable extension for SPARTA.

---

### Official Review · AnonReviewer2 · 2020-11-04
**Borderline paper. Need proofreading.**

**Rating:** 5
**Confidence:** 3

**Review:**


1. Summary:

This paper address efficient searching technique in partially observed MDP. The proposed a learned belief search (LBS) and use auto-regression model to learn the belief distribution of unobserved information. With the learned belief model, they can estimate the expected return via VDN technique. They evaluate the method on Hanabi and obtain 60\% benefit of the exact search with $35\times$ reduction in computation cost.
However, it seems that there are many typos or technique issues, i think this paper needs proofreading before it can be accepted.

2. Some Concerns/weakness:

(1) When using argmax of $Q(a^i|\tau^i)$ in n-step rollout, how to handle the overestimation? especially, there is a large variance in rollout, do you use any variance reduction technique?

(2) All the experiments are evaluated on Hanabi. Does this method can solve different imperfect information games? I am not sure whether the improvement are specially designed for this particular game. I want to see at least one experiment evaluated on another different game, such as Leduc.

(3) This paper is not well written. There are many typos in the equation or some other paragraphs, which i will list in the following.

3. Questions:

(1) what's BP in "when the BP was trained via RL.", page 1. BP is not defined beffor it's used.

(2) what's blueprint policies in section 2.1?

(3) the belief is not well defined in section 3. “We defined beliefs $B^i(\tau_t)=P((s_t, \{\tau^j_t\})|\tau^i_t)$, which is the probability distribution ...”.  What's $\{\tau^j_t\}$?

(4) In Eq.2, it's a little bit confusing. in the expectation of $R^t(\tau')$, $t$ refers to the horizon index of $\tau'$?

(5) could you explain equation 3 and 4 , they are very important in this paper.


4. some issues/typos:

[1] page 2, "Simplified Action Decoder (SAD) (?)"

[2] We denote the environment trajectory as $\tau_{t}=\{s_0, a_0, ..., s_t, a_t\}$

[3] page 8, "At the heart of LBS is an autoregressive model", remove "at"

[4] page 8, "This could e.g. be addressed by retraining", remove "e.g."

[5] page 1, " the policies of any other agents is available at test time", is -> are.

[6] MC rollouts refers to Monte Carlo rollouts?

---

> ### Author Response · Authors · 2020-11-20
> **Response**
>
> Thanks for the detailed review. We have fixed all the typos mentioned in the review in our revision.
>
> **Regarding the concern/weakness:**
>
> (1) Yes, overestimation is a problem for small number of MC samples, although this problem disappears when the number of Monte Carlo samples goes to infinity. SPARTA did two tricks to allow for smaller number of samples. First, it uses an upper-confidence-bound (UBC) for sampling actions. Therefore, overestimation will lead to increased sampling for that given action which will reduce the variance. Secondly, the final estimate has to exceed the blueprint performance by at least a threshold, delta, for us to deviate from the blueprint. Delta is chose such that this is unlikely to happen based on variance. LBS inherit those tricks from SPARTA. On top of that LBS is also more sample-efficient because the bootstrapped rollouts have lower variance.
>
> (2) LBS is designed for Dec-POMDPs, i.e. fully cooperative, partially observable settings. Leduc poker is a zero-sum game and thus out of scope for our method. To the best of our knowledge Hanabi is the only large scale game that is fully cooperative / partially observable and has been established as an AI benchmark.
>
> **Regarding the questions:**
>
> (1) & (2) BP means blueprint, and blueprint policy refers to the underlying policy on top of which the search or other policy improvement techniques are applied.
>
> (3) $\tau_{t_{j}}$ is action observation historys (AOH) of all players. In Dec-POMDPs the beliefs must model other agents' AOHs as well as the current state, since in general the policies of other players condition on these AOHs. We have also added the explanation to the revised paper.
>
> (4) $R^{t}(\tau’)$ means the expected total return from the starting time $t$ until the end of the game.
>
> (5) Eq. 3 says that the exact belief over partners’ private observations (i.e. the distribution over partner private observation histories given my observation history) can be approximated by a neural network model parameterized by $phi$. The private observations can further be decomposed into a sequence of tokens which can be modeled in an auto-regressive way.
> Eq. 4 says that LBS can take advantage of the Q-values to “bootstrap” the total future reward rather than computing the estimate of future rewards entirely through a Monte Carlo rollout. We split the sum, $R_t =  \sum_{t’ = t}^{t’ = T} r_t’$, into two parts. The first n-terms a sampled from the rollout, while the rest of the terms are estimated via the Q-function, $R_t  \approx \sum_{t’=t}^{t'=t+n-1} r_{t}’ + Q_{VDN}(s_{t+n})$. Here s_t is shorthand for the entire state of the game, including the public history and the private observations of all players. We have also updated the paper to include more explanation for Eq. 3 and Eq.4.
>
> **Regarding the issues:**
>
> Thanks again for pointing out the errors. We fixed them in our revision.
>
> Regarding (8): Yes MC rollouts mean Monte Carlo rollouts. We have also updated it in the paper.

---

### Official Review · AnonReviewer5 · 2020-11-06

**Rating:** 5
**Confidence:** 1

**Review:**

**Summary**
This paper is an extension of previous work SPARTA, with two improvements: one is to use a learned belief model to sample in a large state space; the other is to improve the efficiency by replacing full rollouts with partial rollouts that bootstrap from a value function after a specific number of steps. Experiments on the target game, Hanabi shows its superior performance than previous belief search methods in terms of efficiency.

**Strengths**
This paper is more scalable by learning an auto-regressive belief model via supervised learning.

**Weaknesses**
LBS only supports one agent in a two-player game.

**Detailed comments**
I do not have sufficient expertise to comment on LBS (sections 3 and 4).

**Some questions**
One question is that the authors mention learning a centralized reward function using VDN, while other agent’s q-value functions are unavailable. Also, I don’t understand how to use the total reward expectation to avoid having to unroll episodes until the end of the game.

**Small comments**
Some notations are not explained well. For example, on page 1, what does BP mean? BP is not given the full name. It is hard to understand without domain knowledge.
On page 3, there is a typo about the reference of SAD, with a symbol ‘?’

---

> ### Author Response · Authors · 2020-11-20
> **Response**
>
> Many thanks for the feedbacks! We have fixed the notation issues in the revision. BP is defined in Section 3 on page 3 “blueprint (BP)”. We have moved the definition to page 1. We have also fixed the reference of SAD.
>
> Regarding your main question:
>
> All agents’ observations and Q values are available during centralized training, but not at test time. In cooperative games, VDN trains a joint value function $Q_{VDN} = Q^{1} + Q^{2}$, which gives a more accurate estimate of the expected future return than independent Q-learning. LBS combines VDN with a new public-LSTM architecture, in which all players share the same public history. Therefore, for computing the Q-value contributions from each of the players at test time the only missing piece is their private observation, which we sample from the belief model.
>
> Given this sampled full state of the game, the next step is to estimate the total future return for each of the possible actions. In SPARTA this was done by rolling out the episode until the end, following the blueprint. This is both costly and can lead to high variance.  In contrast, LBS takes advantage of the Q-values to “bootstrap” the total future reward. This is very similar to what is done in n-step Q-learning (https://arxiv.org/pdf/1602.01783.pdf) or GAE (https://arxiv.org/abs/1506.02438). Rather than computing the estimate of future rewards entirely through a Monte Carlo rollout, we instead split the sum, $R_t =  \sum_{t’ = t}^{T} r_{t’}$, into two parts. The first n-terms a sampled from the rollout, while the rest of the terms are estimated via the Q-function,  $R_t  \approx \sum_{t’=t}^{t+n-1} r_{t’} + Q_{VDN}(s_{t+n})$. Here $s_t$ is shorthand for the entire state of the game, including the public history and the private observations of all players.

---

### Decision · Program_Chairs · 2021-01-07
**Final Decision**

**Decision:**

Reject

**Comment:**

This paper has some interesting ideas and is an incremental improvement over previous work. However, it needs further revisions and polishing. The relation to prior work is a bit unclear. Since you mention POMDPs, what would be an equivalent version of your method in POMDPS? Why not compare your algorithm with a state-of-the-art method for small discrete problems? It is also a bit unclear why training a model to predict beliefs would be faster than just calculating them (after all the data must come from somewhere)..